# Riehl’s Melanosis: A Multimodality, In Vivo, Real-Time Skin Imaging Study with Cellular Resolution Optical Coherence Tomography and Advanced Skin Diagnosis System in a Tertiary Medical Center

**DOI:** 10.3390/bioengineering9090419

**Published:** 2022-08-26

**Authors:** Peng-Chieh Shen, Yu-Pei Chan, Chun-Hsien Huang, Chau Yee Ng

**Affiliations:** 1School of Medicine, College of Medicine, Chang Gung University, Taoyuan 33302, Taiwan; 2Department of Dermatology, Chang Gung Memorial Hospital, Taoyuan 33305, Taiwan; 3Vitiligo Clinic and Pigment Research Center, Chang Gung Memorial Hospital, Linkou District, New Taipei 33305, Taiwan; 4Department of Dermatology and Aesthetic Medicine Center, Jen-Ai Hospital, Taichung 41265, Taiwan

**Keywords:** Riehl’s melanosis, dermoscopic, skin imaging analysis, optical coherence tomography

## Abstract

Background: Riehl’s melanosis is a psychologically devastating hyperpigmentary disorder that typically occurs on the face and neck. The study of Riehl’s melanosis is limited due to its rarity, variable morphology, and lack of noninvasive diagnostic tools. Recent advances in skin imaging analysis and diagnostic systems improve diagnostic accuracy and enable the noninvasive, real-time evaluation of pigmentary disease. A comprehensive study of Riehl’s melanosis clinical morphology with multimodality and in vivo skin imaging systems has yet to be reported. Objectives: To investigate the clinical features and in vivo advanced skin imaging findings of Riehl’s melanosis. Methods: We retrospectively investigated the clinical characteristics, dermoscopic, and histopathological features of Riehl’s melanosis. We further utilized multimodality skin imaging analysis systems, including a cellular resolution optical coherence tomography (OCT) and new skin diagnosis system, to investigate the features of Riehl’s melanosis. In addition, we compared OCT findings with histopathological features and clinical assessment. Results: We evaluated 30 patients with Riehl’s melanosis at a tertiary medical center from 2010 to 2022. The average age was 47.7 ± 12.3 (mean ± SD) years, predominantly female patients (female: *n* = 23; male: *n* = 7). Cellular resolution OCT imaging from lesion skin shows increased melanocyte capping, disrupted basement membrane, telangiectatic blood vessels, and melanophages in the dermis. The advanced skin diagnosis system captured subclinical erythema of the skin, highlighting the inflammatory nature of the disease. The results correlated well with histopathological findings. Limitations: This is a single-center, cross-sectional study. Conclusions: We highlight the features of Riehl’s melanosis through a novel cellular resolution OCT and photographic skin diagnosis system. A multimodality skin diagnosis system can serve as a real-time, in vivo, noninvasive method for evaluating pigmentary disorders.

## 1. Introduction

Riehl’s melanosis, first reported by Riehl in 1917 [1], is a pigmented contact dermatitis that presents with brown to bluish-gray ill-defined hyperpigmentation over the face and neck [2]. The pigmentation is more pronounced on the forehead, neck, and lateral face, including the zygomatic and temporal region, with relative sparing in the central part [3]. It is believed to be primarily due to contact or photocontact sensitivity caused by certain cosmetics’ fragrances, preservatives, or bactericides [4]. Thus, a patch or photopatch test may be helpful to screen for possible causative factors. The pathogenesis of Riehl’s melanosis remains uncertain. Recently, the immune reaction caused by intrinsic or extrinsic factors and the role of paracrine melanogenic molecules has been postulated [5,6]. 

To date, the diagnostic criteria of Riehl’s melanosis have not been well established. Riehl’s melanosis often demonstrated variability of clinical manifestation. The hyperpigmentation in reported cases had different colors ranging from gray, brown, grayish-brown, reddish-brown, to bluish-brown [7], making visual assessment quite challenging. Thus, an objective, noninvasive, and reliable skin imaging analysis with reproducible outcomes is fundamental. In pigmentary disorders, these assessment techniques include polarized light photography, colorimetry, diffuse reflectance spectroscopy, hyperspectral imaging, and reflectance confocal microscopy [8]. Optical coherence tomography (OCT) is a novel technique with increased popularity [9]. It uses low-power infrared laser light and can produce high-contrast images of skin with good resolutions [10]. The technology has been used to study various dermatological disorders like basal cell carcinoma [9,11], scleroderma [10], psoriasis vulgaris [12], and onychomycosis [13]. To our knowledge, this is the first study that applied OCT to Riehl’s melanosis. 

Riehl’s melanosis is a crucial issue in a patient with skin color. However, research in this field is limited, and most reports are small case series in Japan, India, Korea, and China. In this study, we comprehensively investigate the clinical, dermoscopic, and histopathological features of a comparable number of patients with Riehl’s melanosis in a tertiary medical center in Taiwan. We first utilized a novel, multimodality, in vivo, real-time skin imaging analysis, including cellular resolution OCT and skin diagnosis system, to study the features of Riehl’s melanosis.

## 2. Materials and Methods

### 2.1. Study Design

We retrospectively collected the electronic medical records, clinical images, dermoscopic images, pathological reports, and OCT data of patients who visited Chang Gung Memorial Hospital, Taiwan, between 2010 and 2022. Patients were recruited based on the clinical and histopathological diagnosis of Riehl’s melanosis. Clinical data such as age, sex, skin lesion’s location, possible causative agents, onset and duration of the disease, and associated symptoms such as inflammation, pruritus, erythema, and scaling were recorded. Patients with incomplete data or poor image quality were excluded. The institutional review board (IRB) at our hospital approved this study (IRB no: 202200591B.0) 

### 2.2. Conventional Clinical and Dermoscopic Imaging

The clinical images were obtained by a Nikon digital camera D90 (Nikon, Tokyo, Japan). Wood’s lamp (Dermlite Lumio2) emits light from 320 nm to 450 nm and distinguishes epidermal and dermal melanin. The dermoscopic images were acquired by camera coupling with a dermoscopy (HEINE Optotechnik, Herrsching, Germany). A contact polarized mode with ultrasound gel for immersion was utilized. Based on previously published works [4,14], six features, including pseudo network, gray dots or granules, telangiectatic vessels, perifollicular whitish halo, tiny scales, and follicular keratotic plugs, were evaluated. The dermoscopic findings were defined as present if all authors reached a consensus.

### 2.3. Histopathological Analysis

Based on analysis of hematoxylin and eosin-stained skin tissue slices, the diagnosis of Riehl’s melanosis was made independently by at least two pathologists. The pathologists discussed the discrepancies with the dermatologists to reach a final consensus.

### 2.4. Multimodality Advanced Skin Imaging System

In three patients, we utilized the novel OBSERV 520× skin diagnosis device and a cellular level, high-resolution optical coherence tomography skin imaging system (ApolloVue S100, Apollo Medical Optics, Ltd., Taipei, Taiwan) to examine the features of Riehl’s melanosis. We further analyzed the findings of skin diagnosis device and cellular resolution OCT compared with clinical and histopathological features of the twenty-four cases.

The OBSERV 520× skin diagnosis system (OBSERV^®^, Eindhoven, The Netherlands) is equipped with observation modes, including erythema, parallel-polarized, cross-polarized mode, and a UV mode that can quantify the extent of involved skin. Erythema mode illustrates the network of microvascular structures in the skin leading to redness and facial flushing. Parallel-polarized mode is an enhanced view of skin surface details such as fine lines, wrinkles, texture, and pores. On the other hand, the cross-polarized mode suppresses surface shine for an unobstructed view of the dermal structure, vascular conditions, and pigmentations. Ultraviolet (UV) mode enhances the visualization of epidermal melanin. 

The OCT system used a Ti: sapphire crystal fiber as a light source and elliptical-shape illumination [14,15]. Conventional OCT had a low resolution, with an axial resolution of around 10 μm and lateral resolution of 7.5 μm [16]. The ApolloVue S100 consists of a B-scan, and en-face scan imaging of human skin shows a vertical image, including the epidermis, dermis, and dermal–epidermal junction (DEJ) with cellular resolution simultaneously. The field of views (FOVs) of B-scan and en-face images were 500 × 400 μm^2^ (imaging depth) and 500 × 500 μm^2^, respectively [17]. This cellular resolution full-field (FF) OCT has an axial resolution of 1.5 μm and a lateral resolution of 1.1 μm [18]. 

Since lesions of Riehl’s melanosis were mottled with hyper and hypopigmented areas, three images were taken from each patient for comparison: hyperpigmented lesional skin, hypopigmented lesional skin, and normal skin. All authors reviewed the digital images. 

### 2.5. Statistical Analysis

Categorical variables are reported as numbers (percentage). Statistical analysis was performed with SPSS version 24.0 software for Windows (IBM, Armonk, NY, USA) and GraphPad Prism version 9.0 software for Windows (San Diego, CA, USA). 

## 3. Results

### 3.1. Clinical Features

We collected a roster of 30 patients, 23 (76.7%) female and 7 male (23.3%), from the year 2010–2022. Detailed demographic data were summarized in Table 1. The average onset age was 47.7 ± 12.3 (mean ± SD) years old, and the average disease duration before visiting a dermatologist was 19.6 ± 26.4 months. Common sites of involvement were reported as the face (90%) and neck (63.3%). Associated symptoms included inflammation (16.7%), pruritus (53.3%), erythema (33.3%), and scaling (10.0%). No patient complained of pain or burning sensation. Of 30 patients, 14 (46.7%) had prior exposure to cosmetics, followed by perfumes and essential oil in 2 cases and toner products and sunscreens in 1 case, respectively. A patient had undergone a patch test. The result disclosed a 1+ reaction to nickel. 

### 3.2. Conventional Clinical and Dermoscopic Imaging

Clinically, a reticular confluent pigmentation on the face and neck could be observed under room light. Wood’s light examination shows enhancement in certain areas (Figure 1). Dermoscopic images reveal features such as pseudonetwork (100%), gray dots/granules (100%) telangiectatic vessels (73.3%), perifollicular whitish halo (60%), slight floury scales (50.0%), and follicular keratotic plugs (20.0%). All Riehl’s melanosis lesions displayed at least three dermoscopic features, whereas 60.0% had more than four features, 30.0% had more than five features, and 16.7% of lesions presented with all six dermoscopic signs. The median number of dermoscopic features appearing in a single lesion was four (Figure 2).

### 3.3. Histopathological Characteristics

Among 30 patients, 24 cases had undergone skin biopsy and pathology studies. Key histopathological features on hematoxylin and eosin stain were assessed. Significant findings of Riehl’s melanosis were dermal inflammation (87.5%), perivascular inflammatory cells infiltration (79.2%), melanophages in dermis (62.5%), basal layer degeneration (58.3%), interface change (54.2%) and pigmentary incontinence (54.2%) (Figure 3). A direct immunofluorescence study was examined in three cases, all of which revealed negative results.

### 3.4. Multimodality Advanced Skin Imaging System 

The skin diagnosis system (OBSERV^®^ 520×) revealed reticular brown to gray skin hyperpigmentation under room light. Cross-polarized light decreases surface reflectance and enhances pigmentation visualization. UV light accentuated the pigmentary regions and displayed distinctive mottled hyperpigmentation and hypopigmentation features. Erythema mode disclosed mild redness and increased microvascular structures over the lesional skin (Figure 4).

In OCT images of patients with Riehl’s melanosis, we observed clusters of cap-like hyper-reflective pigmentations, known as melanin capping. It represented keratinocytes containing abundant melanin on top of their nuclei. Spotty, donut shape, and hyper-reflective melanophages were found in the dermis. The blurred DEJ indicated basal keratinocytes vacuolar degeneration. Irregular acanthosis and vessel dilatation with increased blood flow were signs of dermal inflammation (Figure 5a). The hypopigmented lesional skin showed nearly no melanin capping nor dermal melanophages. It resembled normal skin based on our OCT studies (Figure 5b).

## 4. Discussion

Riehl’s melanosis, a variant of noneczematous contact dermatitis, predominantly affects mid-aged females of Asian descent. Patients with Riehl’s melanosis often have significantly impaired quality of life [19] because of the slight improvement despite long-term treatment [20]. Although it is classified as a type IV allergic reaction due to a repeat small amount of contact sensitization to the skin immune system [21], causative factors cannot be easily discovered or avoided. Our study and previous works demonstrated that patch test results and predisposing factors were probably inconsistent [2,22]. The clinical manifestations of Riehl’s melanosis are diverse in patterns and colors of pigmentation [7] and overlap in morphology with other pigmentary disorders [23]. Up to now, the diagnosis of Riehl’s melanosis is still very challenging, and the diagnostic criteria have not been established. The histopathological examination has good value in the diagnosis of Riehl’s melanosis. The most significant findings of Riehl’s melanosis are vacuolar degeneration of the basal layer and pigment incontinence of the dermis [2,4]. It could help us differentiate from other pigmentary disorders like melasma, lichen planus pigmentosus, exogenous ochronosis, and nevus of Ota [23,24,25]. However, a skin biopsy is needed before the pathology exam. Patients are often reluctant to receive facial biopsies because of the discomfort, pain, and the risk of bleeding, infection, or scarring causing cosmetic disfigurement, a remarkable number of patients underwent biopsy in our study because of the persistent lesion after struggling with long-term treatment. Hence, it is essential to develop other diagnostic tools which allow us to have a histopathological view noninvasively.

The characteristics of noninvasive tools are summarized in Table 2. Compared with conventional dermoscopy and room light observation, the novel advanced skin diagnosis system and cellular resolution OCT demonstrate to be able to show the details of the skin, in high accordance with the histopathological features. Cross-polarized light decreases the skin reflectance of the surface skin and enables a more precise assessment of the extent of hyperpigmentation. UV light is absorbed by melanin; therefore, it accentuates the color variation between pigmented and normal skin. Mild redness under the erythema mode may correlate with telangiectatic vessels, suggesting a role of subclinical inflammation in Riehl’s melanosis. In sum, these varying lights from the powerful skin analyzing system lower the visual detection threshold and highlight the areas of concern. 

OCT produces noninvasive, real-time images that resemble histopathological features [13]. It has excellent application prospects for its high-resolution, broader imaging fields, and deeper imaging depths. OCT can produce a longitudinal view of the tissue, similar to a traditional histopathological picture. In normal skin under OCT, the stratum corneum was composed of a thin upper hyper-reflective layer and a gray-black zone underneath. Then, there was a thin layer of hyper-reflective cells suggesting stratum granulosum, followed by a grayish thicker stratum spinosum layer with round, small, hyporeflective keratinocytes. Melanin was revealed as hyper-reflective small spots in the basal layer. The dermal–epidermal junction (DEJ) could separate the epidermis and dermis. The dermal area showed a heterogeneous pattern with dark linear blood vessels embedded (Figure 6). Prescriptively, wood’s light is used to speculate whether the pigment deposition is predominantly epidermal, dermal, or mixed [26]. However, its effectiveness is limited in patients with dark skin [27]. With the help of OCT, we are able to recognize the site of pigments directly. The skin architecture, including the dermal–epidermal junction, dermal papillae, vasculature, and epidermal thickness [11], can also be measured without biopsy. This study first describes OCT features in Riehl’s melanosis. Melanin capping and melanophages in dermis are assumed to be associated with pigmentary incontinence, and blurred DEJ is supposed to be related to basal layer degeneration, histologically. Although the pathomechanism of Riehl’s melanosis is still uncertain, it is presumed to be associated with exogenous stimuli like fragrance or chemicals, which results in inflammation and further causes basal layer vacuolar degeneration. While it occurs, increased melanocytes’ activities by elevated expression of stem cell factor and c-kit pathway lead to the development of melanogenesis in human skin [5]. The melanins enter the dermis via the damaged DEJ and are phagocytized by macrophages, forming melanophages, shown as spotty, donut shape, hyper-reflective cells under OCT. Melanin capping or supranuclear capping is a cluster of cap-like hyper-reflective pigmented globules containing melanosomes released from dendrites of melanocytes [28]. Keratinocytes capture the globules through the microvilli and ingest melanosomes into the cytosol via the shedding vesicle system. This process is called shedding phagocytosis [28,29]. Then the melanins scatter in the supranuclear area of the keratinocytes and protect the nucleus from UV or other stimulants-induced DNA damages [29]. The features of OCT reflect the pathohistological changes in Riehl’s melanosis.

Compared with other hyperpigmented disorders such as melasma, Riehls’ melanosis is more complicated and requires more time to treat [20]. Several regimens have been attempted with diverse outcomes in small population studies or case reports [20,30,31,32,33,34,35,36,37]. However, there are no standard treatment guidelines established yet. In this study, we proved that OCT is a feasible option for pigmentary disorders. OCT provides real-time vertical images resembling conventional pathological appearances without the need for biopsy. It is an excellent application to evaluate treatment response and sequential follow-up since subtle dermal inflammation, melanocytes, and melanin pigments can be observed with clarity.

The limitation of this study is that it was performed in a single health system with a limited sample size. The maximum scanning depth of the cellular resolution FFOCT was 400 μm, limiting the scanning depth in the upper dermis. Lesions involving deep dermis and subcutis cannot be evaluated well. The use of OCT also requires training to be skillful. Nonetheless, this is the first study that comprehensively analyzes Riehl’s melanosis with a multimodality skin imaging approach, including a cellular resolution OCT. OCT is a promising new method for objectively monitoring pigmented skin disease and brings images with good resolution, high magnification, and is familiar to both dermatologists and pathologists [12,15,38].

In conclusion, the features of Riehl’s melanosis through a novel cellular resolution OCT have been demonstrated. The multimodality skin diagnosis system can serve as a better approach in comparison with the traditional method and is of considerable value in evaluating pigmentary disorders [9,11,39].

## 5. Conclusions

Multimodal skin imaging helps evaluate pigmentary disorders and can provide reliable and objective measurements. We demonstrated a comprehensive study of Riehl’s melanosis with skin diagnostic imaging and novel high-resolution OCT analysis.

## Figures and Tables

**Figure 1 bioengineering-09-00419-f001:**
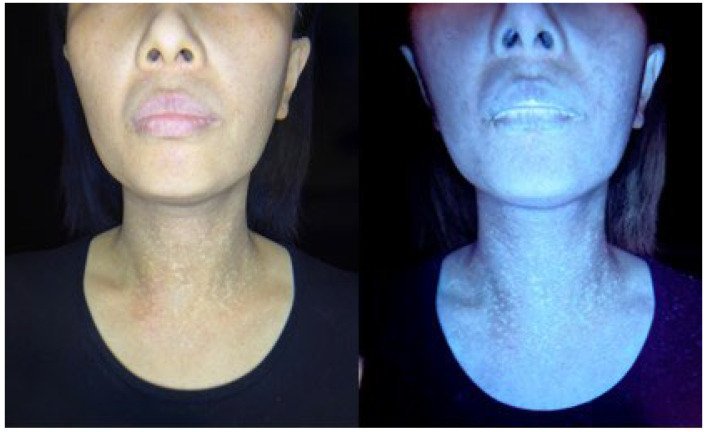
Mottled hyperpigmentation and hypopigmentation on the neck are distinctive features in patients with Riehl’s melanosis. Wood’s lamp examination showed accentuation of macules. (Left: Room light, right: Wood’s lamp, Dermlite Lumio2).

**Figure 2 bioengineering-09-00419-f002:**
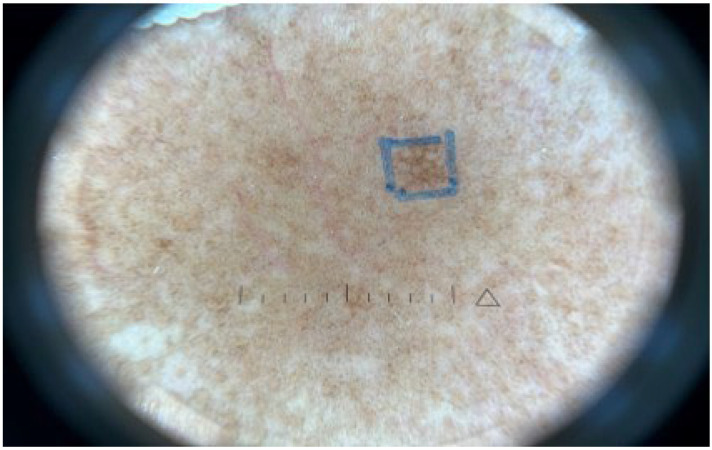
Pseudonetwork, gray dots/granules, telangiectatic vessels, perifollicular whitish halo, fine scales, and follicular keratotic plugs are features of Riehl’s melanosis under dermoscopy.

**Figure 3 bioengineering-09-00419-f003:**
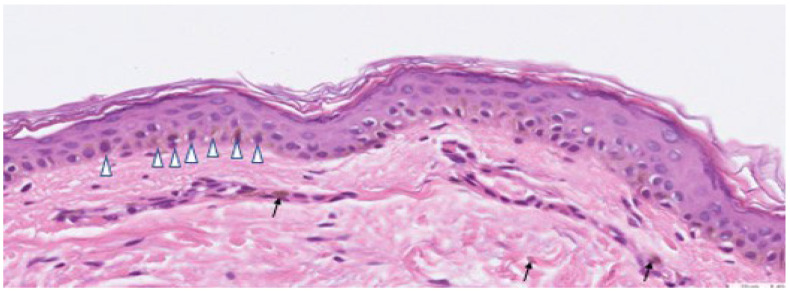
Histopathology features of Riehl’s melanosis showed basal cell vacuolization with necrotic keratinocytes, prominent pigment incontinence, melanin supranuclear cap (shown in 
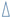
), and melanophages in the dermis (shown in arrow).

**Figure 4 bioengineering-09-00419-f004:**
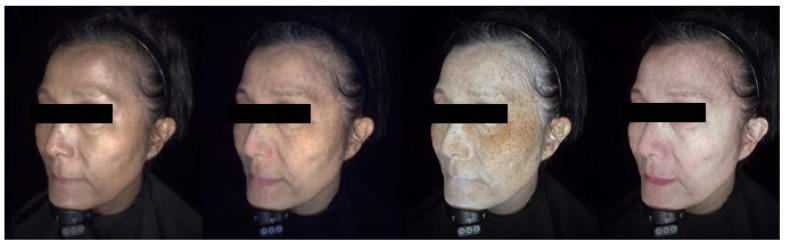
A 50-year-old female experienced progressively darkening skin after applying essential oil for two years. The skin diagnosis system reveals irregular brown to gray hyperpigmentation over cheeks and forehead under room light. Cross-polarized light enhances the hyperpigmented area. True UV light significantly highlights the dyschromic regions. Erythema can be observed over lesional skin, suggesting subclinical inflammation of the skin. (From left to right: room light mode; cross-polarized light mode, UV mode, erythema mode, OBSERV^®^ 520× skin diagnosis system).

**Figure 5 bioengineering-09-00419-f005:**
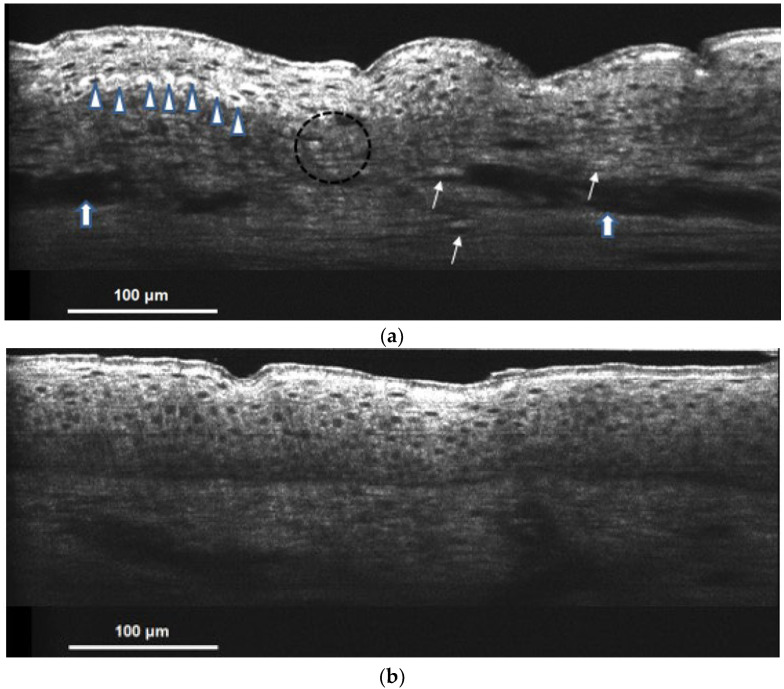
(**a**) Hyperpigmented lesional skin of Riehl’s melanosis on cellular resolution OCT. Hyper-reflective supranuclear capping is presented (shown in 
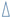
). Spotty donut shape and hyper-reflective melanophages can also be observed in the dermis (shown in the arrow). Blurred DEJ represents basal keratinocyte degeneration (shown in a circle). Irregular acanthosis and vessel dilatation with increased blood flow (shown in 
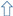
) are the signs of dermal inflammation; (**b**) hypopigmented lesional skin of Riehl’s melanosis on cellular resolution OCT showed nearly no melanin capping nor dermal melanophages and resembled normal skin.

**Figure 6 bioengineering-09-00419-f006:**
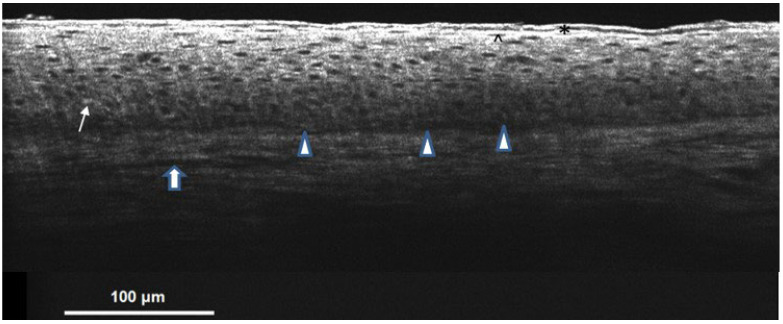
Normal skin on cellular resolution OCT. The stratum corneum is composed of a thin upper hyper-reflective layer and a gray-black zone underneath (shown in *). Then, there is a thin layer of hyper-reflective cells suggesting stratum granulosum (shown in ^), followed by a grayish thicker stratum spinosum layer with round, small, hyporeflective keratinocytes. The hyper-reflective melanin (shown in the arrow) is intermittently presented in the basal cell layer. The epidermis and dermis separated by the dermal–epidermal junction can be clearly illustrated (shown in 
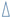
). The dermal area showed a heterogeneous pattern with dark linear blood vessels (shown in 
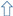
) embedded.

**Table 1 bioengineering-09-00419-t001:** Demographic data of patients with Riehl’s melanosis.

Characteristics	Value
No. of cases	30
Sex, *n* (%)	
Male	7 (23.3)
Female	23 (76.7)
Age at onset (yrs), *n* (%)	47.7, 12.3 (Mean, SD)
Disease duration (mo)	19.6, 26.4 (Mean, SD)
Site of Riehl’s melanosis, *n* (%)	
Face	27 (90.0)
Neck	19 (63.3)
Associated symptoms, *n* (%)	
Inflammation	5 (16.7)
Pruritus	16 (53.3)
Erythema	10 (33.3)
Scaling	3 (10.0)
No. of cases with contact history (Causative factors), *n* (%)	15 (50.0)
Cosmetics	14
Essential oil	2
Perfume	2
Sunscreens	1
Toner product	1
Treatment, *n* (%)	
Topical medications (including hydroquinone cream, retinoid cream, azelaic acid, and steroid cream)	30 (100.0)
Oral medications (including tranexamic acid, antihistamine, and ascorbic acid)	14 (46.7)
Laser and light device Nd-YAG	8 (26.7) (including 2 Nd-YAG laser, 2 IPL, 3 Picolaser, and 1 Ruby laser)

**Table 2 bioengineering-09-00419-t002:** Multimodality features of Riehl’s Melanosis with high-resolution OCT, skin diagnosis system, dermoscopy, and histology.

Diagnostic Tools	Characteristics
Cellular resolution OCT	
Hyperpigmented Lesional Skin	- Clusters of cap-like hyper-reflective pigmentations were presented. - Spotty, donut shape, hyper-reflective melanophages were found in the dermis - Dermo-epidermal junction blurred resembling liquefaction degenerating - Irregular acanthosis - Vessel dilatation with increased blood flow indicates dermal inflammation.
Hypopigmented Lesional Skin	- Resemble normal tissue
Skin Diagnosis System	- Cross-polarized mode eliminates surface reflection and delineates pigmentation and erythema area - UV light and wood’s light accentuate the mottled, reticular hyper-and hypopigmentation - Erythema mode displays mild redness over the lesional skin
Dermoscopy	- Pseudonetwork, gray dots and granules, telangiectatic vessels, perifollicular whitish halo, fine scales, and follicular keratotic plugs
Histology	- Dermal inflammation, perivascular inflammatory cell infiltration, epidermal inflammatory cell infiltration, melanophages in the dermis, pigmentary incontinence, basal layer degeneration, and interface change by infiltration of lymphocytes, mononuclear cells, and eosinophils

## Data Availability

Not applicable.

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
