# Peer review of "Riehl’s Melanosis: A Multimodality, In Vivo, Real-Time Skin Imaging Study with Cellular Resolution Optical Coherence Tomography and Advanced Skin Diagnosis System in a Tertiary Medical Center"

_bioengineering, 2022, doi:10.3390/bioengineering9090419_

Round 1

Reviewer 1 Report

An interesting study about the use of advanced diagnostic devices in the management of Rhiel's melanosis. 

A paragraph describing how this novel device may impact the treatment of this dermatosis, and a review of currently available treatments, are, in my opinion, mandatory before publication

Thank You

Author Response

We appreciate the careful and thoughtful reviews provided with the opportunity to resubmit a revised manuscript. We have revised the article according to your and other reviewer’s advice. We provide a point-by-point response below to address the comments from the editor as requested. The enclosed manuscript file has all changes highlighted in red text. We appreciate your consideration of our work and hope you will deem these revisions acceptable.

Point-to-point reply to the editor’s comments:  

Specific comments and corrections: 

 Reviewer 1
1. A paragraph describing how this novel device may impact the treatment of this dermatosis, and a review of currently available treatments, are, in my opinion, mandatory before publication

Response: We’ve added the paragraph as “Compared with other hyperpigmented disorders such as melasma, Riehls’ mela-nosis is more complicated and requires more time to treat[20]. Several regimens have been attempted with diverse outcomes in small population studies or case reports[20, 30-37].  However, there are no standard treatment guidelines established yet. In this study, we proved that OCT is a feasible option for pigmentary disorders. OCT provides real-time vertical images resemble conventional pathological appearances without the need for biopsy. It is an excellent application to evaluate treatment response and sequential follow-up since subtle dermal inflammation, melanocytes and melanin pigments can be observed with clarity.”

Page 11, Paragraph 1, Line 273-281

Compared with other hyperpigmented disorders such as melasma, Riehls’ mela-nosis is more complicated and requires more time to treat[20]. Several regimens have been attempted with diverse outcomes in small population studies or case reports[20, 30-37].  However, there are no standard treatment guidelines established yet. In this study, we proved that OCT is a feasible option for pigmentary disorders. OCT provides real-time vertical images resemble conventional pathological appearances without the need for biopsy. It is an excellent application to evaluate treatment response and sequential follow-up since subtle dermal inflammation, melanocytes and melanin pigments can be observed with clarity.

Reviewer 2 Report

The aim of this paper, entitled “Riehl’s melanosis: a multimodality, in-vivo, real-time skin imaging study with cellular resolution optical coherence tomography and advanced skin diagnosis system in a tertiary medical center” is to investigate the features of Riehl’s melanosis through a multimodal skin imaging analysis systems, including a cellular resolution optical coherence tomography (OCT).

The possible application of new technologies to dermatological diagnosis is certainly an interesting topic, however the work presents some criticisms.

First, the purpose of this work is not entirely clear, and the conclusions do not seem adequately supportedRiehl melanosis is considered form of pigmented allergic contact dermatitis, typically to fragrance and other ingredients of cosmetic products. Even if specific dermoscopic and OCT aspects have been described, the diagnosis is mainly clinic-anamnestic and requires the execution of allergy tests to identify and remove the possible causative agent. Beyond the descriptive value, the investigation methodology proposed by the authors, therefore, has scarce clinical applicability.

Furthermore, the authors do not seem to describe characteristic patterns that allow a clear identification of the pathology, even considering its differential diagnoses

Authors collected the cases presented over 12 years, between 2010 and 2022. Was the equipment used always the same? The technical characteristics of equipments are implemented very frequently, and this can lead to results that are not easily comparable.

The number of patients undergoing biopsy is remarkably high (24/30). Is this a routine approach for patients suspected of Riehl's melanosis or were the biopsies done for research purposes only?

Minor points:

The initial paragraph of the Results section 3.4. (Optical Coherence Tomography Findingsdoes not report results but general considerations, which should be better incorporated in the Introduction or Discussion.

The discussion is not consistent and does not allow to fully understand what the logic of the work is, in addition to the descriptive one. For example: why were the biopsies performed and how do the histological findings integrate with the other observations?

Author Response

We appreciate the careful and thoughtful reviews provided with the opportunity to resubmit a revised manuscript. We have revised the article according to your and other reviewer’s advice. We provide a point-by-point response below to address the comments from the editor as requested. The enclosed manuscript file has all changes highlighted in red text. We appreciate your consideration of our work and hope you will deem these revisions acceptable.

Point-to-point reply to the editor’s comments:  

Specific comments and corrections: 

Reviewer 2

  1. First, the purpose of this work is not entirely clear, and the conclusions do not seem adequately supported. Riehl melanosis is considered a form of pigmented allergic contact dermatitis, typically to fragrance and other ingredients of cosmetic products. Even if specific dermoscopic and OCT aspects have been described, the diagnosis is mainly clinic-anamnestic and requires the execution of allergy tests to identify and remove the possible causative agent. Beyond the descriptive value, the investigation methodology proposed by the authors, therefore, has scarce clinical applicability.

Response: The purpose of our study to compare the multimodality advance skin analysis including the skin diagnosis system (OBSERV® 520X) and a novel cellular resolution OCT to conventional diagnostic methods. The skin diagnosis system possess varying lights which are able to lower the visual detection threshold and highlight the areas of concern. OCT produces non-invasive, real-time images that resemble histopathological features. These new techniques provide non-invasive, real-time evaluation with greater details of the skin, in high accordance with the histopathology when comparing with conventional approach (observe under room lights, Wood’s lamp and dermoscopy). The contexts were revise in the conclusion as “In conclusion, the features of Riehl’s melanosis through a novel cellular resolution OCT have been demonstrated. The multimodality skin diagnosis system can serve as a better approach in comparison with traditional method and is of considerable value in evaluating pigmentary disorders[9, 11, 39] ”

Page 8, Paragraph 1, Line 219-222

Compared with conventional dermoscopy and room light observation, the novel ad-vanced skin diagnosis system and cellular resolution OCT demonstrate to be able to show the details of the skin, in high accordance with the histopathological features.

Page 11, Paragraph 3, Line 291-294

In conclusion, the features of Riehl’s melanosis through a novel cellular resolution OCT have been demonstrated. The multimodality skin diagnosis system can serve as a better approach in comparison with traditional method and is of considerable value in evaluating pigmentary disorders[9, 11, 39].  

  1. Furthermore, the authors do not seem to describe characteristic patterns that allow a clear identification of the pathology, even considering its differential diagnoses

Response: We’ve added the sentence as “The histopathological examination has good value in the diagnosis of Riehl’s melano-sis.. The most significant findings of Riehl's melanosis are vacuolar degeneration of the basal layer and pigment incontinence of the dermis[2, 4]. It could help us differentiate with other pigmentary disorders like melasma, lichen planus pigmentosus, exogenous ochronosis, and nevus of Ota[23-25].”

Page 7, Paragraph 1, Line 209-213

The histopathological examination has good value in the diagnosis of Riehl’s melano-sis.. The most significant findings of Riehl's melanosis are vacuolar degeneration of the basal layer and pigment incontinence of the dermis[2, 4]. It could help us differentiate with other pigmentary disorders like melasma, lichen planus pigmentosus, exogenous ochronosis, and nevus of Ota[23-25].

  1. Authors collected the cases presented over 12 years, between 2010 and 2022. Was the equipment used always the same? The technical characteristics of equipments are implemented very frequently, and this can lead to results that are not easily comparable.

Response: Our study is a retrospective study, out of thirty patients, twenty four had obtained histopathological feature. All thirty patients underwent room light photography, Wood’s lamp and dermoscopy features (as written in Section 2.2. and 3.2) “Convential clinical and dermoscopic imaging”). Multimodality advanced skin diagnostic tool utilizing the skin diagnosis system (OBSERV® 520X) and a novel high resolution OCT device (ApolloVue S100, Apollo Medical Optics, Ltd., Taipei, Taiwan) was used to evaluate the features in three patients. We further compare these findings to histopathological features to prove the feasibility of these new diagnositic device as compare to conventional diagnositic methods. (Section 2.4 and 3.4 “Multimodality advanced skin imaging system” ). We have revised the article accordingly.

Page 1, Abstract, Line 25-26

We retrospectively investigated the clinical characteristics, dermoscopic, and histopathological features of Riehl’s melanosis.

Page 2, Paragraph 3, Line 73-75

We retrospectively collected the electronic medical records, clinical images, der-moscopic images, pathological reports, and OCT data of patients who visited Chang Gung Memorial Hospital, Taiwan, between 2010 and 2022.

Page 2, Paragraph 4, Line 81-90

2.2. Conventional clinical and dermoscopic imaging

The clinical images were obtained by a Nikon digital camera D90 (Nikon, Tokyo, Japan). Wood’s lamp (Dermlite Lumio2) emits light from 320 nm to 450 nm and dis-tinguishes epidermal and dermal melanin. The dermoscopic images were acquired by camera coupling with a dermoscopy (HEINE Optotechnik, Herrsching, Germany). A contact polarized mode with ultrasound gel for immersion was utilized. Based on pre-viously published works [4, 14], six features, including pseudo network, gray dots or granules, telangiectatic vessels, perifollicular whitish halo, tiny scales, and follicular keratotic plugs, were evaluated. The dermoscopic findings were defined as present if all authors achieved a consensus.

Page 3, Paragraph 2-5, Line 96-121

2.4. Multimodality advanced skin imaging system

In three patients, we utilized the novel OBSERV 520x skin diagnosis device and a cellular level, high-resolution optical coherence tomography skin imaging system (ApolloVue S100, Apollo Medical Optics, Ltd., Taipei, Taiwan) to examine the features of Riehl’s melanosis. We further analyzed the findings of skin diagnosis device and cellular resolution OCT compared with clinical and histopathological features of the twenty four cases.

OBSERV 520x skin diagnosis system (OBSERV®, Eindhoven, Netherlands), equipped with observation modes including erythema, parallel-polarized, cross-polarized mode, and UV mode that can quantify the extent of involved skin. Er-ythema mode illustrates the network of microvascular structures in the skin leading to redness and facial flushing. Parallel-polarized mode is an enhanced view of skin sur-face details such as fine lines, wrinkles, texture, and pores. On the other hand, the cross-polarized mode suppresses surface shine for an unobstructed view of the dermal structure, vascular conditions, and pigmentations. UV (Ultraviolet) mode enhances the visualization of epidermal melanin.

The OCT system used a Ti: sapphire crystal fiber as a light source and ellipti-cal-shape illumination[14, 15]. Conventional OCT had a low resolution, with an axial resolution of around 10 μm and lateral resolution of 7.5 μm[16]. The ApolloVue S100 consists of a B-scan, and en face scan image of human skin shows a vertical image in-cluding the epidermis, dermis, and dermal-epidermal junction (DEJ) with cellular res-olution simultaneously. The field of views (FOV) of B-scan and en face images were 500 × 400μm2 (imaging depth) and 500 × 500 μm2, respectively[17]. This cellular reso-lution full-field (FF) OCT has an axial resolution of 1.5 μm and a lateral resolution of 1.1 μm[18]. 

Since lesions of Riehl’s melanosis were mottled with hyper and hypopigmented areas, three images were taken from each patient for comparison: hyperpigmented le-sional skin, hypopigmented lesional skin, and normal skin. All authors reviewed the digital images.

Page 5, Paragraph 1, Line 141-150

3.2. Convential clinical and dermoscopic imaging

Clinically, a reticular confluent pigmentation on face and neck could be observed under room light. Wood's light examination shows enhancement in certain areas. Dermoscopic images reveal  features like pseudo-network(100%), gray dots/granules (100%) telangiectatic vessels (73.3%), perifollicular whitish halo (60%), slight floury scales (50.0%), and follicular keratotic plugs (20.0%).

Page 5-6, Paragraph 3, Line 164-170

3.4. Multimodality advanced skin imaging system

The skin diagnosis system (OBSERV® 520X) revealed reticular brown to gray skin hyperpigmentation under room light. Crossed-polarized light decreases surface re-flectance and enhances pigmentation visualization. UV light accentuated the pigmen-tary regions and displayed distinctive mottled hyperpigmentation and hypopigmenta-tion features. Erythema mode disclosed mild redness and increased microvascular structures over the lesional skin (Figures 3).

  1. The number of patients undergoing biopsy is remarkably high (24/30). Is this a routine approach for patients suspected of Riehl's melanosis or were the biopsies done for research purposes only?

Response: We’ve added the sentence as “Patients often reluctant to receive facial biopsies because of the discomfort, pain, and the risk of bleeding, infection, or scarring causing cosmetic disfigurement, a remarka-ble number of patients underwent biopsy in our study because of the persistent lesion after struggling with long-term treatment. Hence, it is essential to develop other diag-nostic tools which allow us to have histopathological view non-invasively.”

Page 7-8, Paragraph 1, Line 215-219

Patients often reluctant to receive facial biopsies because of the discomfort, pain, and the risk of bleeding, infection, or scarring causing cosmetic disfigurement, a remarka-ble number of patients underwent biopsy in our study because of the persistent lesion after struggling with long-term treatment. Hence, it is essential to develop other diag-nostic tools which allow us to have histopathological view non-invasively.

  1. Minor points:

The initial paragraph of the Results section 3.4. (Optical Coherence Tomography Findings) does not report results but general considerations, which should be better incorporated in the Introduction or Discussion.

Response: We’ve incorporated the paragraph in the Discussion

Page 10, Paragraph 1, Line 236-243

In normal skin under OCT, the stratum corneum was composed of a thin upper hy-per-reflective layer and a gray-black zone underneath. Then, there was a thin layer of hyper-reflective cells suggesting stratum granulosum, followed by a grayish thicker stratum spinosum layer with round, small, hypo-reflective keratinocytes. Melanin was revealed as hyper-reflective small spots in the basal layer. The dermal-epidermal junc-tion (DEJ) could separate the epidermis and dermis. The dermal area showed a heter-ogeneous pattern with dark linear blood vessels embedded (Figure 6).

  1. The discussion is not consistent and does not allow to fully understand what the logic of the work is, in addition to the descriptive one. For example: why were the biopsies performed and how do the histological findings integrate with the other observations?

Response: We addressed the above questions as follow (1) why were the biopsies performed (2) how do the histological findings integrate with the other observations. We have adjust the Discussion and put emphasize on the multimodality skin diagnosis system we proposed and the correlation between OCT features and histological findings.

(1) why were the biopsies performed?

Page 7-8, Paragraph 1, Line 210-218

The histopathological examination has good value in the diagnosis of Riehl’s melano-sis.. The most significant findings of Riehl's melanosis are vacuolar degeneration of the basal layer and pigment incontinence of the dermis[2, 4]. It could help us differentiate with other pigmentary disorders like melasma, lichen planus pigmentosus, exogenous ochronosis, and nevus of Ota[23-25]. However, a skin biopsy is needed before the pa-thology exam. Patients often reluctant to receive facial biopsies because of the discom-fort, pain, and the risk of bleeding, infection, or scarring causing cosmetic disfigure-ment, a remarkable number of patients underwent biopsy in our study because of the persistent lesion after struggling with long-term treatment.

(2) how do the histological findings integrate with the other observations.

Page 10, Paragraph 1, Line 248-264

This study first describes OCT features in Riehl’s melanosis. Melanin capping and melanophages in dermis is assumed to be associated with pigmentary incontinence, and blurred DEJ is supposed to be related to basal layer degeneration, histologically. Although the pathomechanism of Riehl’s melanosis is still uncertain, it is presumed to be associated with exogenous stimuli like fragrance or chemicals, which results in in-flammation and further caused basal layer vacuolar degeneration. While it occurs, in-creased melanocytes’ activities by elevated expression of stem cell factor and c-kit pathway lead to the development of melanogenesis in human skin[5]. The melanins enter the dermis via the damaged DEJ and are phagocytized by macrophages, forming melanophages, shown as spotty, donut shape, hyper-reflective cells under OCT. Mela-nin capping or supranuclear capping is clusters of cap-like hyper-reflective pigmented globules containing melanosomes released from dendrites of melanocytes[28]. Keratinocytes capture the globules through the microvilli and ingest melanosomes into the cytosol via the shedding vesicle system. This process is called shedding phagocy-tosis[28, 29]. Then the melanins scatter in the supranuclear area of the keratinocytes and protect the nucleus from UV or other stimulants-induced DNA damages[29]. The features of OCT reflect the patho-histological changes in Riehl’s melanosis
